# How Does Entropy Influence Modern Text-to-SQL Systems?

**Chris Lazar***, **Varun Kausika***, **Satya Saurabh Mishra***, **Saurabh Jha, Priyanka Pathak**
Dell CFS AI Research

## Abstract

In the field of text-to-SQL candidate generation, a critical challenge remains in quantifying and assessing the confidence in the generated SQL queries. Existing approaches often rely on large language models (LLMs) that function as opaque processing units, producing outputs for every input without a mechanism to measure their confidence. Current uncertainty quantification techniques for LLMs do not incorporate domain-specific information. In this study, we introduce the concept of query entropy for Text-to-SQL candidate confidence estimation and integrate it into existing popular self-correction pipelines to guide generations and prevent resource overuse by including a novel clustering technique for generated SQL candidates based on entropy. We further study the treatment of different candidate generation techniques under this paradigm.

## 1 Introduction

Text-to-SQL, the task of converting natural language queries into structured SQL commands, has emerged as a crucial technology in bridging the gap between human users and database systems [Gunther & Beretta (2000); Zhong et al. (2017)]. This field has gained significant attention due to its potential to democratize data access, allowing non-technical users to interact with complex databases using everyday language [Lei et al. (2024); Wang et al. (2019)]. However, the process of accurately translating natural language into SQL queries presents numerous challenges, including the need to understand context, resolve ambiguities, and map diverse linguistic expressions to precise database schema elements [Maamari et al. (2024)].

In recent years, the application of large language models (LLMs) to the text-to-SQL problem has shown promising results [Yu et al. (2019); Lee et al. (2025)]. These models leverage vast amounts of training data to generate SQL queries from natural language inputs. The current state-of-the-art approaches commonly employ the following steps: first, generating a large number of candidate SQL queries, then refining the generated candidates through self-correction [Wu et al. (2024); Song et al. (2025)] and selecting the best candidate [Pourreza et al. (2024); Gao et al. (2024)]. This evolution in methodology has significantly improved the accuracy and robustness of text-to-SQL systems.

Candidate generation methods are common in a multitude of fields apart from text-to-sql [Fredrikson et al. (2015); Hoffmann et al. (2021)]. Despite advances in generation methods, current approaches face limitations that hinder reliability. In particular, existing systems generate a large candidate set without providing confidence scores for the overall set. This lack of uncertainty quantification leads to inefficiencies in the candidate selection process and may result in suboptimal query choices. Furthermore, the absence of a mechanism to determine when to stop refining candidates can cause unnecessary computational overhead [Xia et al. (2024); Cao et al. (2024)].

Inspired by semantic uncertainty [Kuhn et al. (2023); Nikitin et al. (2024)], we attempt to use entropy as a novel measure to quantify uncertainty and guide the refinement of generated SQL queries, thus addressing the aforementioned limitations.

Our contributions are as follows:

---

*Equal Contribution

1. We propose a novel measure for uncertainty and develop a method for clustering similar queries, based on the entropy over their execution result representations.

2. We attempt to incorporate entropy scores into conventional query-fix module, resulting in optimizing resource allocation.

3. We present a cluster analysis of candidate generation methods, offering insights into the distribution of generated SQL candidates within clusters for each method.

## 2 RELATED WORK

**Agentic Frameworks**. Recent state-of-the-art Text-to-SQL frameworks have mainly employed agentic builds, comprising of modules for schema-linking, candidate-generation and selection. These methods rely on supervised fine-tuned language models in conjunction with in-context learning (ICL) for candidate generation and selection. In **XiYan-SQL** [Gao et al. (2024)], a novel hierarchical schema representation has been proposed to highlight relationships between in a semi-structured format. In **CHESS-SQL** [Talaei et al. (2024)], the relevant schema information has been retrieved through a combination of locality sensitive hashing (LSH), semantic similarity and edit distance with the query.

Existing agentic methods mainly place emphasis on selecting majority candidate by providing execution result based clustering result as context to LLMs [Gao et al. (2024); Wang et al. (2019)]. In our work, we explore an alternative entropy based clustering.

**Uncertainty in text-to-sql**. Several studies have been done on uncertainty quantification in LLMs. In Xiong et al. (2024), elicited confidence scores from the model are scored by the model, followed by an aggregation function. In Kadavath et al. (2022), calibration is evaluated based on two tasks: $P(T)$ and $P(IK)$ - the probability that the answer to the question is true and the probability that the LLM knows the answer to the question respectively. In the Text-to-SQL field, The most related work to ours is that of **SUN** [Qin et al. (2022)] which uses token based log probabilities to estimate the intrinsic uncertainty in the candidate SQL queries generated by an LLM. However, unlike our method, it focuses on the uncertainty in the natural language questions themselves and not on the generated candidates.

## 3 PRELIMINARIES

Our method borrows several techniques from **CHASE-SQL** [Pourreza et al. (2024)]. Specifically, we re-use the Information Retrieval, Candidate Generation and Selection Agent modules. The **Information Retrieval (IR)** module identifies relevant tables, columns, values through schema linking and multi-path reasoning strategies. The **Candidate Generator (CG)** utilizes a fine-tuned language model to produce diverse SQL query candidates, ensembling queries from the following techniques for diversity:

1. **Divide and Conquer.** Follows a map-reduce prompting strategy of decomposing the natural language question, answering each question independently and aggregating the results.

2. **Query Plan.** Comprises of breaking the question into a sequence of steps before sub-query generation.

3. **Online Synthetic Example Generation.** A prompting strategy to generate in-context learning examples depending on available table schemas.

Subsequently, the **Query Fixer (QF)** refines candidate queries iteratively by prompting an LLM with the execution result, appropriate schema and few-shot examples. Finally, the **Selection Agent (SA)** pairwise selects the most appropriate query from refined candidates, given schemas as context.

## 4 METHODOLOGY

### 4.1 QUERY ENTROPY

Query entropy quantifies the diversity and uncertainty of generated SQL candidates, serving as a guide for the query refinement process. This metric, detailed in Algorithm 1, is computed on clusters of semantically similar queries. To form these clusters, we leverage execution result embeddings, creating a more nuanced representation of query similarity. Specifically, we convert the execution results into HTML format and utilize the markuplm-base model [Li et al. (2022)] to generate rich, contextual embeddings. Following this, we use DBSCAN clustering [Ester et al. (1996)] to get distributions over candidates. Note that generate embeddings over all candidates, and include their error message in the HTML if they do not run.

---

**Algorithm 1** calculateQueryEntropy

---

  **Input:** candidates, databaseName
  **Input:** eps, minSamples                                   // DBSCAN hyperparameters
  **Initialize:** embeddings $\leftarrow \phi$
  **for** candidate in candidates **do**
    execResult $\leftarrow$ execute(candidate, databaseName)
    HTMLResult $\leftarrow$ HTML(candidate, databaseName)
    embeddings $\leftarrow$ embeddings $\cup$ markupEmbed(HTMLResult)
  **end for**
  clusters $\leftarrow$ DBSCAN(embeddings, eps, minSamples)
  queryEntropy $\leftarrow \sum_{i=1}^{k} p_i log p_i$       // k = number of clusters, $p_i$ = probability of $i^{th}$ cluster
  **return** queryEntropy

---

### 4.2 QUERY FIXER (QF)

The QF module iteratively refines candidate queries to improve correctness and efficiency. Our enhancement lies in using query entropy as a stopping criterion, continuing until the entropy falls below a predetermined threshold relative to the previous generation. This process ensures that we generate high-quality SQL queries while optimizing computational resources. The steps involved are:

- **Entropy-Guided Refinement**: Using query entropy as a stopping criterion, combined with CHASE-SQL's self-debugging approach to iteratively refine candidates by correcting logical or syntactical errors based on execution results or error feedback.

- **Execution-Based Refinement**: Incorporating test-case-driven refinement methods where generated queries are executed on synthetic databases, and errors are used to guide corrections.

## 5 EXPERIMENTS

Our experiments were carried out using Mixtral-8x7b-instruct [Jiang et al. (2024)] for information retrieval and candidate generation, and Deepseek-coder-v2-lite-instruct [DeepSeek-AI et al. (2024)] for query fix and selection agent modules. The models were deployed on three NVIDIA-A100-80GB GPUs. All of our experiments were conducted on a subsampled set of 146 questions from the BIRD benchmark development set. Our results reveal that the optimal number of query-fix iterations is 4 (ref 1.1, 1.2), but the mean Q-ent over iterations remains in the 2.6 - 2.8 range, indicating the need for a more informative clustering approach.

We conducted further experiments on the entropy of the distribution of methods and the coefficient of variation across buckets for different methods (ref 2.1, 2.2), averaged over all questions. Our analysis shows that Divide-and-Conquer(DAC) has the highest entropy, this means DAC is spread out more evenly across the different buckets at each iteration. it doesn't concentrate in a particular bucket but rather distribute more uniformly. Query Plan (QP) and Online Synthetic Example Generation (SYNTH) have the same(lower) entropy, these two methods are less spread out compared to DAC.

Their distributions are more concentrated in fewer buckets. DAC might be a more exploratory method, being used more evenly across different buckets, while QP and SYNTH could be more specialized, being assigned to specific buckets more often. Furthermore, our findings reveal that SYNTH and DAC have the same (higher) coefficient of variation (CV), this possibly means that both the methods fluctuate more in their distributions across iterations. some buckets get significantly higher/lower values over time, leading to more relative variability in percentage allocation. QP has a lower CV, this suggests QP's distribution is more stable over time. it does not fluctuate as much in how its percentages are spread across buckets.

Even though DAC is more evenly distributed according to entropy analysis, it also has high CV, which suggests that distribution is not stable over time, and has fluctuations. SYNTH behaves similarly in terms of variability. QP, on the other hand, has lower entropy and lower CV, meaning it is both more concentrated and more stable in how it spreads across buckets.

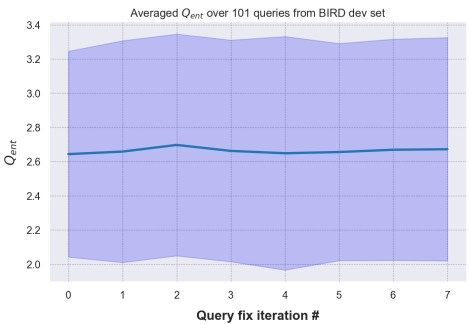

Figure 1.1: Standard deviation in query entropy with query fix iteration, the shaded area corresponds to the standard deviation

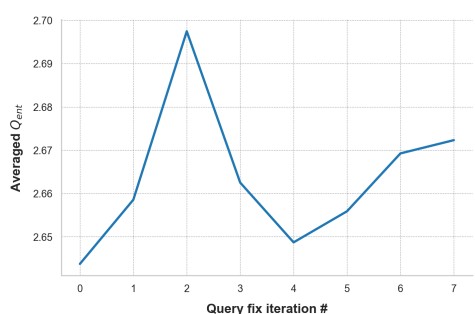

Figure 1.2: Query entropy progression with iterations

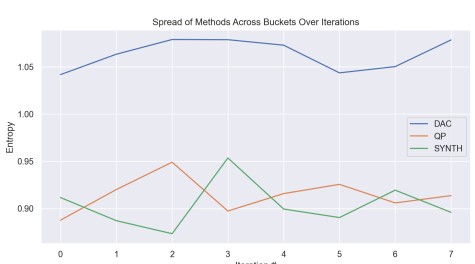

Figure 2.1: Bucket distribution entropy of different candidate generation methods vs. query-fix iteration

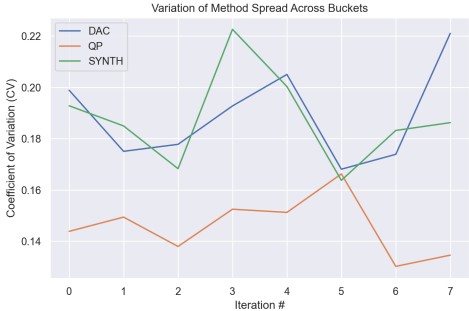

Figure 2.2: Bucket distribution coefficient of variance vs. query-fix iteration

## 6 CONCLUSION

In this paper, we introduced a novel approach to text-to-SQL parsing that integrates query execution plans with semantic clustering techniques. Our method leverages the structural information inherent in SQL queries to enhance the accuracy and efficiency of natural language to SQL translation. Building on our current research, future work will focus on three primary directions: (1) enhancing semantic clustering by more deeply integrating query execution plans, (2) exploring specialized language model architectures tailored for SQL generation tasks, and (3) expanding evaluation across broader and more diverse database benchmarks. Specifically, we aim to develop more sophisticated cost-based weighting strategies, investigate adaptive techniques for improving open-source model performance, and incorporate schemas from financial, scientific, and domain-specific databases. Our analysis reveals key findings that will guide these developments, including the observation that query plan techniques demonstrate lower covariance variation compared to traditional approaches and that Query Fix iterations exhibit diminishing returns. By pursuing these research directions, we

aim to push the boundaries of text-to-SQL parsing, ultimately creating more accurate, efficient, and versatile natural language interfaces for complex database querying.

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
