# OpenReview forum: "How Does Entropy Influence Modern Text-to-SQL Systems?"
_ICLR.cc/2025/Workshop/BuildingTrust — BuildingTrust_

### Official Review · Reviewer_9kd3 · 2025-02-19
**Good point**

**Rating:** 6
**Confidence:** 3

**Review:**

Weakness:
No Explicit Trustworthiness Discussion

While entropy is implicitly related to safety and robustness, if the paper does not explicitly connect entropy to trustworthiness concerns, it might seem too abstract for this workshop.
Suggestion: Add sections discussing entropy’s role in LLM safety & ethical AI, such as:
How entropy-aware responses reduce hallucination risks,
Using entropy to detect adversarial prompts and data poisoning attacks.

Strengths:
Strong Theoretical Foundation

Entropy is widely used in calibration, Bayesian uncertainty modeling, and AI safety, making it a scientifically rigorous approach. The paper is smart to use the concept.

---

### Official Review · Reviewer_eaZQ · 2025-03-01
**This paper presents a novel entropy-based approach to uncertainty estimation in text-to-SQL systems, offering a promising method for query refinement.**

**Rating:** 7
**Confidence:** 3

**Review:**

## Strengths

* The paper introduces query entropy as a unique confidence metric for SQL candidate selection, addressing a critical gap in text-to-SQL systems that traditionally lack meaningful uncertainty quantification.

* The use of multiple candidate generation methods (Divide and Conquer, Query Plan, and Synthetic Example Generation) provides a solid empirical basis for evaluating entropy’s role in SQL refinement.

* The adoption of DBSCAN clustering and execution result embeddings via MarkupLM ensures a nuanced approach to grouping SQL queries while optimizing resource use.

* The entropy analysis of different query generation methods (DAC, QP, SYNTH) offers valuable insights into their stability and diversity, contributing to broader discussions on text-to-SQL methodology.

## Weaknesses

* The experiments are conducted on only 146 questions from the BIRD benchmark, which is relatively small for drawing strong generalizable conclusions. The method’s effectiveness on more diverse and complex SQL datasets is uncertain

* The stopping criterion for query refinement based on entropy reduction lacks an adaptive mechanism, meaning it may not generalize well across different SQL tasks or database schemas.

* The paper does not compare query entropy against existing confidence scoring methods (e.g., token log probabilities or calibration techniques), leaving it unclear whether entropy is truly superior.

---

### Decision · Program_Chairs · 2025-03-04

**Decision:**

Accept

**Comment:**

The use of Entropy to estimate the confidence of generated SQL candidates is novel. Good paper overall.